# Multidetector Computed Tomography with Dedicated Protocol for Breast Cancer Locoregional Staging: Feasibility Study

**DOI:** 10.3390/diagnostics10070479

**Published:** 2020-07-14

**Authors:** Vinicius C. Felipe, Luciana Graziano, Paula N. V. P. Barbosa, Vinicius F. Calsavara, Almir G. V. Bitencourt

**Affiliations:** 1Department of Imaging, A.C.Camargo Cancer Center, São Paulo 01509-010, Brazil; vinicius.felipe@accamargo.org.br (V.C.F.); luciana.graziano@accamargo.org.br (L.G.); paula.pinto@accamargo.org.br (P.N.V.P.B.); 2Department of Epidemiology and Statistics, A.C.Camargo Cancer Center, São Paulo 01509-010, Brazil; vinicius.calsavara@accamargo.org.br

**Keywords:** breast neoplasms, neoplasm staging, computed tomography, magnetic resonance imaging

## Abstract

Background: The aim of this study was to demonstrate the feasibility of performing multidetector computed tomography (MDCT) with a dedicated protocol for locoregional staging in breast cancer patients. Methods: This prospective single-center study included newly diagnosed breast cancer patients submitted to contrast-enhanced chest MDCT and breast magnetic resonance imaging (MRI). MDCT was performed in prone position and using subtraction techniques. Fleiss’ Kappa coefficient (K) and intraclass correlation coefficient (ICC) were used to assess agreement between MRI, MDCT, and pathology, when available. Results: Thirty-three patients were included (mean age: 47 years). Breast MRI and MDCT showed at least substantial agreement for evaluation of tumor extension (k = 0.674), presence of multifocality (k = 0.669), multicentricity (k = 0.857), nipple invasion (k = 1.000), skin invasion (k = 0.872), and suspicious level I axillary lymph nodes (k = 0.613). MDCT showed higher number of suspicious axillary lymph nodes than MRI, especially on levels II and III. Both methods had similar correlation with tumor size (MRI ICC: 0.807; *p* = 0.008 vs. MDCT ICC: 0.750; *p* = 0.020) and T staging (k = 0.699) on pathology. Conclusions: MDCT with dedicated breast protocol is feasible and showed substantial agreement with MRI features in stage II or III breast cancer patients. This method could potentially allow one-step locoregional and systemic staging, reducing costs and improving logistics for these patients.

## 1. Introduction

Prognosis of breast cancer depends on the extent of the disease (staging) and biological characteristics of the tumor [1]. Currently, the basic evaluation for locoregional staging of breast cancer patients includes clinical examination, mammography, and ultrasound [2]. Breast magnetic resonance imaging (MRI) has been increasingly used, as it has proven to be superior to conventional imaging methods in assessing tumor extent and identifying additional lesions [3,4]. MRI is able to identify additional tumor foci in 20% of patients on the same breast and 5.5% on the contralateral breast, modifying treatment in up to one third of patients with breast cancer [5]. However, the widespread implementation of this method is limited by its high cost, long exam time, low availability in some regions, and perceived low specificity [6]. Despite more accurate staging with MRI, guidelines differ widely in their recommendations for the performance of pretreatment breast MRI for all newly diagnosed breast cancer patients [3,7].

According to the National Comprehensive Cancer Network (NCCN) guidelines, breast cancer patients classified as stage II or higher should be considered to have diagnostic chest computed tomography (CT) performed, with contrast as part of the staging for distant metastasis [8]. Despite having a high spatial resolution, multidetector CT (MDCT) is considered to have a low accuracy in characterizing breast lesions, which is related to the lower soft tissue contrast resolution when compared to MRI. However, contrast resolution of x-ray imaging for breast evaluation is well-known based on mammography and digital breast tomosynthesis (DBT). The main difference between DBT and CT is that the first is a limited-angle tomographic technique, while the second is a 360° angle technique, providing full three-dimensional capability. Recently, dedicated CT devices developed for breast assessment, using technologies such as “cone-beam” and “phase-contrast”, have shown promising results [9,10]. Moreover, the use of iodinated contrast media on contrast-enhanced mammography (CEM), which are the same media used on CT, has proven to be as effective as the paramagnetic contrast used on MRI to characterize tumor vascularity. However, few studies have evaluated the performance of conventional MDCT devices in the evaluation of breast lesions [11,12,13,14,15,16,17,18]. Compared to MRI, the advantages of CT include lower cost and shorter exam time, in addition to the ability to identify distant metastases in the same exam.

For a proper evaluation of the breast parenchyma in MDCT, we proposed to use a dedicated protocol, to be performed in prone position, similar to breast MRI, in addition to using subtraction techniques, which allow improvement of the contrast resolution between breast cancer and normal tissues, without significant increase in examination time or radiation dose. The aim of this study was to demonstrate the feasibility of MDCT with a dedicated protocol for locoregional staging in breast cancer patients, in comparison to breast MRI and histopathological results.

## 2. Materials and Methods

### 2.1. Patient Selection

This prospective unicentric study included newly diagnosed breast cancer patients submitted to contrast enhanced MDCT for systemic staging and breast magnetic resonance imaging (MRI) for locoregional staging between March 2019 and March 2020, according to the institution’s protocol. Included patients were submitted to chest CT using a dedicated protocol for breast evaluation, and the results were compared to breast MRI and histopathological results. Patients with any contraindications to iodine contrast or MRI were excluded. The study protocol was approved by the institutional ethics review board (approval code 2621/18, approved on 22 January 2019), and all patients gave written informed consent before inclusion.

### 2.2. Histopathologic Analysis

In all patients, breast cancer was diagnosed by percutaneous core needle biopsy. Expression of estrogen receptor (ER), progesterone receptor (PR), and human epidermal growth factor receptor-2 (HER2) were obtained from immunohistochemical (IHC) analysis from pretreatment needle biopsies, according to the 2018 ASCO/CAP guidelines [19,20]. All pathologic results from outside biopsies were reviewed at our institution. Tumors were classified based on IHC results in one of the following molecular subtypes: Luminal (ER/PR-positive; HER2-negative or positive); HER2 overexpressing (ER/PR-negative and HER2-positive); and triple-negative (ER/PR-negative and HER2-negative).

### 2.3. Imaging Acquisition

MDCT was performed with a specific protocol dedicated to breast evaluation, in prone position, using a specially made device (Figure 1), which reproduces the breast MRI coil. Exams were performed on a 160-slice MDCT scanner (Canon Aquilion Prime 160; Canon Medical Systems, Otawara, Japan), with a 0.5 mm slice thickness, before and after administration of 1–2 mL of the nonionic contrast material ioversol (Optiray 320; Mallinckrodt Medical Inc., St. Louis, MI, USA) per kilogram of body weight intravenously with a semiautomated power injector at a rate of 4 mL/s. The acquisition of chest images was performed 80–90 s after the contrast administration, which was previously reported as the optimal delay time to depict breast cancer [21]. Pre-contrast images were digitally subtracted from post-contrast images using a motion correction software (Sure Subtraction; Canon Medical Systems, Otawara, Japan) to improve the evaluation of enhancing lesions in the breast parenchyma and color-coded iodine maps were performed (Figure 2).

Breast MRI was performed in prone position with a 1.5T MR imaging system (Achieva, Philips Healthcare, Best, Netherlands; or Magnetom Aera, Siemens Healthcare, Erlangen, Germany) with an 8-channel dedicated breast coil. A standard dose (0.1 mmol/kg body weight) of gadopentetate dimeglumine (Magnevist; Bayer HealthCare Pharmaceuticals, Wayne, NJ, USA) was injected intravenously as a bolus at 4 mL/s followed by a saline flush. The total MRI examination time was approximately 25–30 min. Breast MRI imaging protocol included:Axial T1 gradient-echo phase, three-dimensional (3D) imaging (TR/TE, 545/9.7 ms; 3-mm-thick slices; FOV, 340 mm).Fat-saturated short tau inversion recovery (STIR) sequence in the sagittal plane of both breasts (TR/TE, 3200/63 ms; 3-mm-thick slices; FOV, 210 mm).Axial diffusion-weighted images (DWI) using spin-echo, single-shot echo planar imaging sequence (TR/TE, 10630/57 ms; 2.5-mm-thick slices; FOV, 350 mm; b-values: 0 and 750 s/mm^2^).Dynamic contrast enhancement (DCE): Five gradient-echo phases in T1, 3D, and in the axial plane using fat suppression for dynamic examination (TR/TE, 3.23/1.6 ms; 0.9-mm-thick slices; FOV, 360 mm), one pre-contrast and four post-contrast with a temporal resolution of 60–90 s.Sagittal T1-weighted, 3D gradient-echo pulse sequence with fat signal suppression (TR/TE, 4.58/2.28 ms; 1-mm-thick slices; FOV, 220 mm).

### 2.4. Image Analysis

MDCTs were evaluated by two radiologists with 4 years’ and 10 years’ experience in breast and cancer imaging who were blind to breast MRI findings. Breast MRI examinations were independently analyzed by another two radiologists with 4 to 15 years’ experience in breast imaging. For both imaging methods, conventional imaging (mammography and ultrasound) as well as histological results from prior percutaneous biopsy were available, and discordant cases were reviewed in consensus between readers.

Both breast MRI and MDCT findings were classified according to the 5th edition Breast Imaging Reporting and Data System (BI-RADS^®^) MRI-lexicon [22]. The index tumor was classified according to the imaging phenotype as mass, non-mass enhancement (NME), or both (mass and NME), and its size was measured on the largest diameter at any plane. The lesions were also classified as unifocal, multifocal, or multicentric, and the disease extent was measured including all tumor foci. Multifocality was defined as additional sites of malignancy within the same breast quadrant of the index tumor, while multicentricity was defined as additional sites of malignancy within different quadrants of the same breast. Additional lesions identified on breast MRI or MDCT that led to additional benign biopsies were also described. Direct invasion of skin, nipple, or chest wall, as well as the number and location of suspicious axillary lymph nodes were assessed in both imaging methods. The morphologic criteria to define an axillary lymph node as suspicious on both MDCT and MRI were cortical thickening, absence of fatty hilum, and round or oval shape. Average tumor density on MDCT (HU, Hounsfield units) was calculated on pre- and post-contrast images, using a region of interest (ROI) that covered most of the lesion.

### 2.5. Statistical Analysis

Frequencies and percentages were used to describe categorical variables, and mean, median, standard-deviation (SD), and ranges were used to describe continuous variables. MDCT features were compared with MRI findings and histological results. Differences in tumor size and extension smaller than or equal to 10 mm between breast MRI and MDCT were considered concordant, while differences greater than 10 mm were considered discordant. Bland–Altman plots and intraclass correlation coefficient (ICC) with 95% confidence interval (95% CI) were used to assess agreement on tumor extent between MRI, MDCT, and pathology when available. ICC values less than 0.5, between 0.5 and 0.75, between 0.75 and 0.9, and greater than 0.90 were considered as poor, moderate, good, and excellent agreement, respectively [23]. Fleiss’ Kappa coefficient (K) with standard error (SE) was used to assess agreement between breast MRI and MDCT features. Agreement was considered poor (K less than 0.19), weak (K between 0.20 and 0.39), moderate (K between 0.40 and 0.59), substantial (K between 0.60 and 0.79), or almost perfect (K higher than 0.80) [24]. The significance level was fixed at 5% for all tests, that is, p values less than 0.05 were considered to represent statistically significant results. All analyses were performed using SPSS software, version 20.0 (SPSS Inc., Chicago, IL, USA) and R software, version 3.5 (R Foundation for Statistical Computing, Vienna, Austria).

## 3. Results

Thirty-three patients were included in this study. Mean age was 47 years (median: 45 years; SD: 10.4 years; range: 31–74 years). Most common histologic type was no special type (NST) invasive ductal carcinoma (*n* = 29; 87.9%) and most tumors had nuclear grade III (*n* = 28; 84.8%). Regarding molecular subtypes, 21 (63.7%) of the tumors were classified as luminal, 9 (27.3%) as triple-negative, and 3 (9.1%) as HER2 overexpressing.

The main tumor was identified on MRI and MDCT in all cases. Table 1 summarizes imaging findings on both modalities and agreement between them. At breast MRI, mean tumor extension was 4.6 cm (median: 3.9 cm; SD: 3.3 cm; range: 1.4–12.2 cm). At MDCT, mean tumor extension was 4.5 cm (median: 3.2 cm; SD: 3.2 cm; range: 1.3–13.0 cm). There was a good agreement on tumor extent between these methods (ICC: 0.868; 95% CI: 0.732–0.935; *p* < 0.001), and both were considered concordant in 28 cases, of which 21 had less than 6 mm difference in tumor extent. Compared to MRI, MDCT overestimated tumor extension in one case (3.0%) and underestimated tumor extension on four cases (12.1%). Multicentricity was characterized in both methods in 10 patients (30.3%) (Figure 3), and only in MDCT in one case (3.0%). There were no contralateral malignant lesions in our sample. In four cases (12.1%), there were suspicious axillary lymph nodes on MDCT, which were not identified on MRI (three were confirmed as malignant and one negative for malignancy after biopsy). In eight cases (24.2%), MDTC showed a higher number of suspicious axillary lymph nodes when compared to MRI, including level II and/or III in five cases.

The mean density of the index tumor on MDCT was 35 HU (median: 35 HU; SD: 8 HU; range: 20–52 HU) on pre-contrast images and 86 HU (median: 87 HU; SD: 19 HU; range: 53–139 HU) on post-contrast images. The mean enhancement was 51 HU (median: 54 HU; SD: 17 HU; range: 25–109 HU). Three patients showed additional findings on MDCT: one with suspicious lung nodules and a sternal lesion (bone lesion also characterized on breast MRI); one with a lytic bone lesion in the sternum (not characterized on breast MRI); and one with diffuse lytic bone lesions (not characterized on breast MRI).

Eleven patients were submitted to surgery before systemic treatment and had the pathological results (gold-standard) compared to breast MRI and MDCT. The analysis of ICC and Bland–Altman plots (Figure 4) showed that MRI had a slightly better agreement with tumor size on pathology (ICC: 0.807; 95% CI: 0.282–0.948; *p* = 0.008) than MDCT (ICC: 0.750; 95% CI: 0.070–0.933; *p* = 0.020); however, both methods showed similar agreement with pathology for T staging (Table 2). Eight cases had metastatic axillary lymph nodes on pathology, including five cases with suspicious lymph nodes on MDCT and three cases with suspicious lymph nodes on MRI. MDCT also showed a better agreement with pathology on the number of metastatic lymph nodes (Table 3).

## 4. Discussion

Our results showed that MDCT with a dedicated breast protocol can be used for distant and locoregional staging in breast cancer patients. MDCT features showed good agreement with breast MRI, which is considered the most accurate imaging method for preoperative breast cancer locoregional staging. Additionally, MDCT showed a higher number of suspicious axillary lymph nodes than MRI, especially on levels II and III, and a lower number of additional benign biopsies. On patients who had available surgical pathology information, MDCT and MRI showed similar agreement with pathology for T staging, and MDCT showed better agreement on the number of metastatic lymph nodes.

Currently, MDCT has a limited role in breast cancer management. This method is not recommended for breast cancer screening or diagnosis due to the higher radiation dose and lower diagnostic accuracy in comparison to conventional breast imaging methods (mammography, ultrasound, and MRI). However, chest MDCT is already performed for distant staging in many breast cancer patients, and a dedicated protocol for breast evaluation can add additional information for locoregional staging and treatment planning, e.g., tumor extension and axillary lymph node status, with similar exam time and radiation dose. Indeed, MDCT could even overcome some of the perceived limitations of breast MRI, such as nodal evaluation and false-positive additional lesions.

Prior studies have also demonstrated that tumor size accessed on MDCT images has a good correlation with pathological tumor size, providing appropriate information for the determination of adequate surgical margins [11,13]. However, there are concerns about the capability of MDCT to identify the intraductal component of a breast cancer. While some authors have shown that MDCT is effective in detecting ductal carcinoma in situ (DCIS), especially the more aggressive types, other authors have shown that MDCT has a lower sensitivity than MRI [14,17,18]. In the present study, we were not able to compare MDCT and MRI in the detection of intraductal spread due to the small sample size, which should be evaluated in future investigations.

Unlike our protocol, previous studies performed MDCT in the supine position and suggested that it could be an advantage over MRI because it allows better surgical simulation. We chose to perform the exams in the prone position, as it improves the evaluation of the breast parenchyma and allows a better correlation to MRI, in addition to the fact that the breast surgeons at our institution are already used to planning surgery with prone breast MRI. Recently, dual-energy CT was also proposed for locoregional staging of breast cancer. Volterrani et al. evaluated 31 patients with breast cancer and found that dual-energy CT had a good correlation with pathologic analysis on tumor size and cancer distribution (unifocal, multifocal, or multicentric); however, there was no comparison to MRI in that study [25]. In our initial experience, dual-energy CT requires little more radiation dose and has imaging quality similar to subtraction, which is why we prefer the second technique.

The density of breast cancers observed in our sample was similar to those of prior studies, and this information could be useful to characterize additional lesions found on MDCT. Lin et al. evaluated 97 patients with 102 breast lesions and found the optimal cut-offs to differentiate malignant from benign lesions were 32 HU on pre-contrast images (sensitivity, 72%; specificity, 71%; accuracy, 72%), 57 HU on post-contrast enhanced images (sensitivity, 100%; specificity, 83%; accuracy, 89%), and 33 HU enhancement between pre- and post-contrast images (sensitivity, 83%; specificity, 95%; accuracy, 93%) [15].

Our data demonstrated that MDCT depicts more suspicious lymph nodes than MRI, especially on axillary levels II and III. These findings could be explained by the limited ability to obtain a complete visualization of the axilla on breast MRI without dedicated axillary protocol [26]. Prior studies have also demonstrated a high accuracy of MDCT in predicting axillary metastases in breast cancer [27,28,29]. Recently, Chen et al. evaluated 148 breast cancer patients and found that the most important predictors of axillary lymph node metastasis on MDCT were cortical thickness > 3 mm and non-fatty hilum. In that study, MDCT’s sensitivity and specificity for axillary lymph node metastasis prediction based on combined-variated analysis were 85.3% and 87.4%, respectively [27]. MDCT can also be used to distinguish axilla nodal metastasis after neoadjuvant chemotherapy; however, the possibility of false-negative nodal micrometastases is higher, especially in patients with node-positive disease on the pre-treatment MDCT [30].

The results of this study should be considered in the context of its limitations. The small sample size may have hindered some results and have precluded further quantitative analyses. Patients with smaller tumors were not included in this study, as they do not routinely perform staging chest CT at our institution; thus, mean tumor size of our sample was larger than the general breast cancer patients’ population. We used 10 mm as the cut-off value to determine concordance between MDCT and MRI based on prior studies [31,32] and potential clinical relevance; however, this value is arbitrary, and there is no standardized cut off value to determine concordance on tumor size between different methods. Because most stage II or III breast cancer patients are now subjected to neoadjuvant chemotherapy, it is not possible to have a proper comparison of tumor extension or number of affected lymph nodes with pathology (gold-standard) in those cases. Nevertheless, the technique presented here showed promising results for breast cancer locoregional staging, which should be confirmed in future larger studies, in order to define an optimized MDCT protocol in this scenario.

In conclusion, MDCT with dedicated breast protocol is feasible and shows substantial agreement with MRI features in newly diagnosed stage II or III breast cancer patients. This method could potentially be useful when breast MRI is contraindicated or unavailable, allowing locoregional and systemic staging with a single examination, reducing costs and improving logistics for these patients.

## Figures and Tables

**Figure 1 diagnostics-10-00479-f001:**
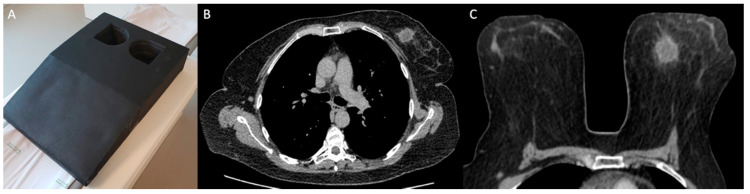
Example of chest multidetector computed tomography (MDCT) with dedicated breast protocol. (**A**) Device used to perform chest MDCT on prone position. (**B**) Axial image of conventional chest MDCT in supine position. (**C**) Axial image of the breast in the same patient using dedicated MDCT protocol in prone position.

**Figure 2 diagnostics-10-00479-f002:**
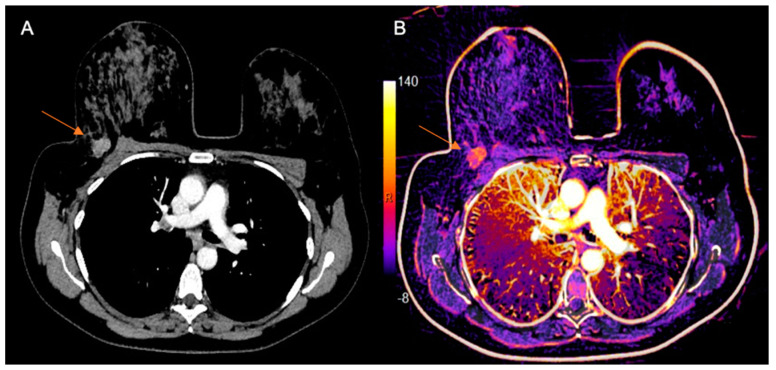
Example of chest MDCT with dedicated breast protocol. (**A**) Post-contrast axial image of chest MDCT in prone position showing a tumor in the right breast (arrow). (**B**) Subtracted imaging with color-coded iodine map showing the same tumor (arrow).

**Figure 3 diagnostics-10-00479-f003:**
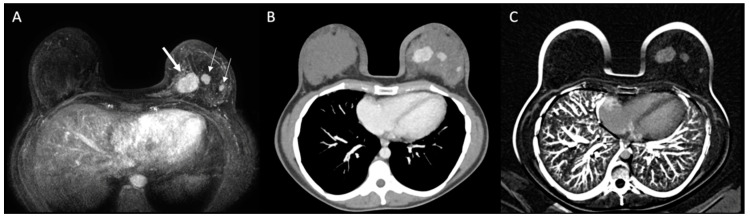
Patient with multicentric invasive breast carcinoma in the left breast. Breast MRI (**A**) identified the main tumor (thick arrow) and two additional lesions on the same breast (thin arrows), which were also identified on MDCT ((**B**): post-contrast; (**C**): subtraction).

**Figure 4 diagnostics-10-00479-f004:**
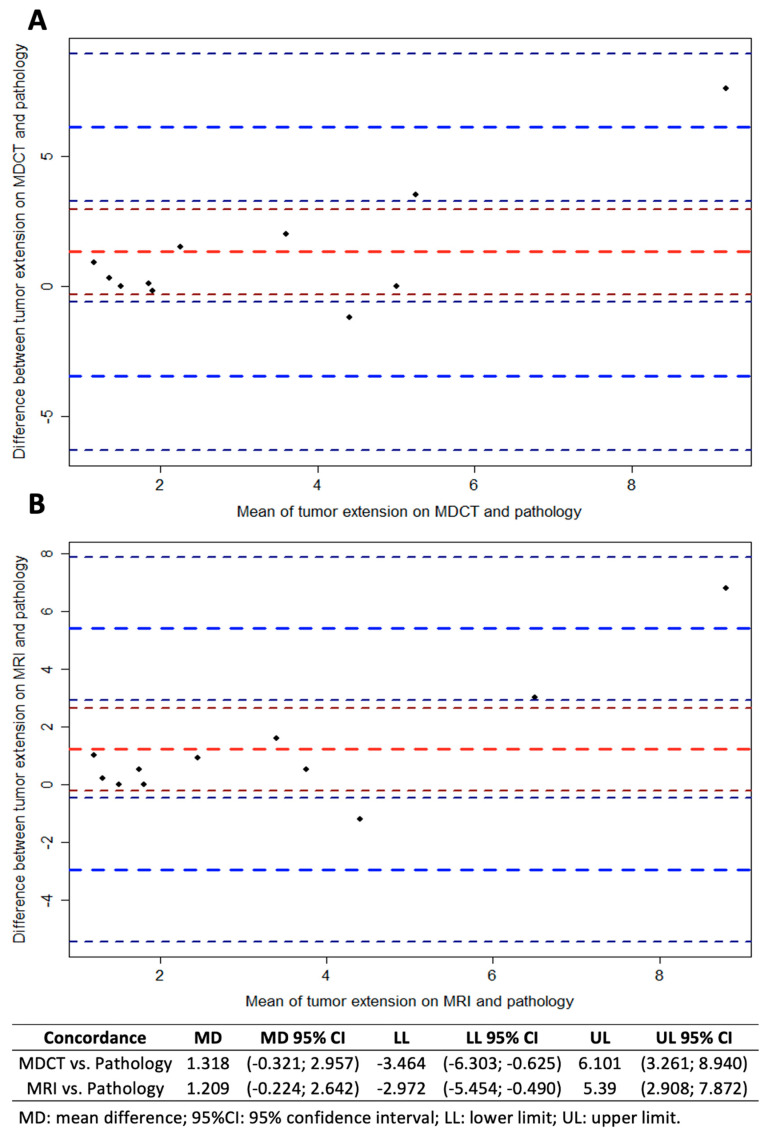
Bland–Altman plots for comparison of the differences between the tumor sizes on MDCT (**A**) or MRI (**B**) and pathology in breast cancer patients (*n* = 11). Red line represents mean differences with 95% CI (dark red lines), blue lines represent the limits of agreement (upper and lower limits) with 95% CI (dark blue lines), and each dot represent an included case.

**Table 1 diagnostics-10-00479-t001:** Imaging findings on breast MRI and MDCT in breast cancer patients (*n* = 33).

Imaging Findings	MR	MDCT	Kappa	SE	*p*
Lesion Type	26 (78.8%)	25 (75.8%)			
Mass		5 (15.2%)	0.756	0.117	<0.001
NME	6 (18.2%)	3 (9.1%)			
Both	1 (3.0%)				
Tumor extension					
≤2 cm (T1)	8 (24.2%)	9 (27.3%)	0.674	0.111	<0.001
2.1–5.0 cm (T2)	14 (42.4%)	15 (45.5%)			
>5 cm (T3)	11 (33.3%)	9 (27.3%)			
Additional benign biopsies	5 (15.2%)	1 (3.0%)	0.298	0.234	0.016
Multifocality	12 (36.4%)	13 (39.4%)	0.678	0.132	<0.001
Multicentricity	10 (30.3%)	11 (33.3%)	0.930	0.069	<0.001
Nipple invasion	2 (6.1%)	2 (6.1%)	1.000	0.000	<0.001
Skin Invasion	4 (12.1%)	5 (15.2%)	0.872	0.125	<0.001
Suspicious axillary lymph nodes					
Level I	19 (57.6%)	23 (69.7%)	0.613	0.138	<0.001
Levels II and/or III	1 (3.0%)	8 (24.2%)	0.178	0.157	0.073
Number of suspected LN					
<3	13 (39.4%)	12 (36.4%)	0.628	0.110	<0.001
≥3	6 (18.2%)	11 (33.3%)			

**Table 2 diagnostics-10-00479-t002:** Comparison between T staging evaluated by breast MRI and MDCT with pathology (*n* = 11).

T Staging on Breast MRI and MDCT	T Staging on Pathology			Kappa (SE)	*p*
	T1	T2	T3		
Breast MRI					
T1 (<2.1 cm)	5	0	0	0.699 (0.185)	0.002
T2 (2.1–5.0 cm)	1 *	3	0		
T3 (>5 cm)	0	1 ^#^	1		
MDCT					
T1 (<2.1 cm)	5	0	0	0.699 (0.185)	0.002
T2 (2.1–5.0 cm)	1 ^†^	3	0		
T3 (>5 cm)	0	1 ^µ^	1		

Discordant cases: * MRI: 2.9 cm; MDCT: 1.8 cm; Pathology: 2.0 cm; ^#^ MRI: 8.0 cm; MDCT: 5.0 cm; Pathology: 5.0 cm; ^†^ MRI: 2.0 cm; MDCT: 3.0 cm; Pathology: 1.5 cm; ^µ^ MRI: 4.0 cm; MDCT: 7.0 cm; Pathology: 3.5 cm.

**Table 3 diagnostics-10-00479-t003:** Comparison of the number of suspected lymph nodes (LN) on breast MRI and MDCT and number of metastatic LN on pathology (*n* = 11).

Number of Suspected LN on Breast MRI and MDCT	Number of Metastatic LN on Pathology			Kappa (SE)	*p*
	0	1–2	>2		
Breast MRI					
0	3	4	1	0.165 (0.152)	0.354
1–2	0	2	1		
>2	0	0	0		
MDCT					
0	3	3	0	0.429 (0.214)	0.038
1–2	0	3	1		
>2	0	0	1		

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
