# Peer review of "Multidetector Computed Tomography with Dedicated Protocol for Breast Cancer Locoregional Staging: Feasibility Study"

_diagnostics, 2020, doi:10.3390/diagnostics10070479_

Round 1

Reviewer 1 Report

Overall I think the paper is well written and describes well what was observed in a clinic for about one year who examined breast cancer patients.

Include how tomosynthesis compares with the proposed methodology – worth including a mention of this technique in the introduction since there is clear connection with MDCT.

Line 120: add space between “to” and “15”

It’s not clear what is represented Figure 4, please improve description of lines, colours, what the dots mean.

This sentence “this method [MRI] is not yet routinely recommended for all breast cancer patients, as it has a high cost, little availability and variable specificity”, is probably too negative when later the following is mentioned: “…breast MRI, which is considered the most accurate imaging method for preoperative breast cancer locoregional staging.”

I’d recommend revisiting the discussion and improving the text for clarity.

Author Response

We would like to thank the reviewer for the comments.

As suggested, tomosynthesis was mentioned in the introduction (page 2, lines 56-59).

A space was added between “to” and “15” on line 120.

Description of the Blant-Altman plots was detailed in Figure 4 legend (page 7, lines 226-228).

The last sentence of the first paragraph in the introduction was rewritten (page 1, lines 41-43).

Reviewer 2 Report

Well written paper on a relevant subject, with good supporting data.

This is a prospective study that investigates the feasibility of multidetector computed tomography (MDCT) with a dedicated protocol for locoregional staging in breast cancer patients.

The major strengths of  the study is the originality, the quality of statistics and the clearly presented results.

Abstract:  adequately represents the manuscript.

Introduction: well written, even if the background could be expanded and references 5 and 6 are not so recent.

Materials and Methods:   this section is well organized. Study protocol is clearly announced, but there is some inaccuracy.

-Page 4 Line 142-144: “Differences in tumor size and extension smaller  than or equal to 10 mm between breast MRI and MDCT were considered concordant, while differences greater than 10 mm were considered discordant.”

I think 10 mm cut off to determine whether the two techniques are concordant or not is too wide.

Results:  Study findings were systematically and clearly announced,  also due to correct and appropriately labeled tables, but there is some weaknesses.

-Page 4 Line 159-163: “At breast MRI, mean tumor extension was  4.6 cm (median: 3.9 cm; SD: 3.3 cm; range: 1.4-12.2 cm). At MDCT, mean tumor extension was 4.5 cm  (median: 3.2 cm; SD: 3.2 cm; range: 1.3-13.0 cm).”

Mean tumor lesion size seems too large, I think this study population doesn’t reflect general breast cancer patients population that routinely needs an accurate preoperative staging.

Discussion:  well written. Limitation of the study were correctly reported.

References : are relevant, but  some of these are not so recent. Reference 17 is incomplete.

In conclusion I suggest to reformulate the aim of the study, focusing on MDCT preoperative staging  role for patients for at > II Stage, because the study doesn’t show clear evidence of  MDCT  feasibility for patients with smaller lesions.

Author Response

We would like to thank the reviewer for the comments and suggestions.

As suggested, more recent references were cited in the introduction (page 2, line 50).

We used 10 mm as the cut off value to determine concordance between MDCT and MRI based in prior studies and potential clinical relevance, however this value in arbitrary and there is no standardized cut off value to determine concordance on tumor size between different methods. This was included as a study limitation (pages 8-9, lines 286-299). In our sample, from the 28 concordant cases (<10 mm difference), 21 had difference < 6 mm. It was also included in the results (page 4, lines 180-181).

We agree that mean tumor size of our sample was larger than the general breast cancer patients’ population. At our institution, patients with smaller tumors (especially < 2 cm) do not usually undergo staging with chest CT. Thus, these patients were not included in the study. This was included as a study limitation (page 8, lines 283-286).

Reference 17 was corrected.

Because cancer stage was not an inclusion criterion in our study, we did not reformulate the study’s objective. However, as most of our patients were stage II or III, we agree with the reviewer comments and included this observation in the conclusion (page 9, line 305).

Round 2

Reviewer 2 Report

I would like to thank you for your reply and for the new version.

Through your insertions in the text you have clarified my doubts and suggestions.

I think that MDCT with dedicated breast protocol could be useful for >stage II breast cancer patient for loco-regional and systemic staging with a single examination. I hope that you will confirm this with further and larger studies.